# Impact of cytokine storm on severity of COVID-19 disease in a private hospital in West Jakarta prior to vaccination

**Diana Laila Ramatillah**[1]*, **Siew Hua Gan**[2], **Ika Pratiwy**[1], **Syed Azhar Syed Sulaiman**[3], **Ammar Ali Saleh Jaber**[4], **Nina Jusnita**[1], **Stefanus Lukas**[1], **Usman Abu Bakar**[3]

**1** Pharmacy Faculty, Universits 17 gustus 1945 Jkrt, North Jakarta, Indonesia, **2** School of Pharmacy, Monash University Malaysia, Bandar Sunway, Selangor, Malaysia, **3** School of Pharmaceutical Sciences, Universiti Sains Malaysia, Penang, Malaysia, **4** Dubai Pharmacy College, UEA, Dubai, United Arab Emirates

* diana.ramatillah@uta45jakarta.ac.id, dianalailaramatillah@gmail.com

## Abstract

### Background and aim

Coronavirus Disease 2019 (COVID-19) has become a worldwide pandemic and is a threat to global health. Patients who experienced cytokine storms tend to have a high mortality rate. However, to date, no study has investigated the impact of cytokine storms.

### Materials and methods

This retrospective cohort study included only COVID-19 positive patients hospitalized in a Private Hospital in West Jakarta between March and September 2020. All patients were not vaccinated during this period and treatment was based on the guidelines by the Ministry of Health Indonesia. A convenience sampling method was used and all patients who met the inclusion criteria were enrolled.

### Results

The clinical outcome of COVID-19 patients following medical therapy was either cured (85.7%) or died (14.3%), with 14.3% patients reported to have cytokine storm, from which 23.1% led to fatalities. A plasma immunoglobulin (Gammaraas®) and/or tocilizumab (inter-leukin-6 receptor antagonist; Actemra®) injection was utilised to treat the cytokine storm while remdesivir and oseltamivir were administered to ameliorate COVID-19. Most (61.5%) patients who experienced the cytokine storm were male; mean age 60 years. Interestingly, all patients who experienced the cytokine storm had hypertension or/ and diabetes compli-cation (100%). Fever, cough and shortness of breath were also the common symptoms (100.0%). Almost all (92.3%) patients with cytokine storm had to be treated in the intensive care unit (ICU). Most (76.9%) patients who had cytokine storm received hydroxychloroquine and all had antibiotics [1) azithromycin + levofloxacin or 2) meropenam for critically ill patients] and vitamins such as vitamins C and B-complex as well as mineral. Unfortunately, from this group, 23.1% patients died while the remaining 70% of patients recovered. A sig-nificant (p<0.05) correlation was established between cytokine storms and age, the

**Data Availability Statement:** All relevant data are available from figshare at https://figshare.com/s/0dc2aaab681c4097f0aa.

**Funding:** The author(s) received no specific funding for this work. Only the funded to held the WCP program not for publication The article is currently not under consideration for publication in any journal. We will be very grateful if our paper is given due respect.

**Competing interests:** The authors have declared that no competing interests exist.

**Abbreviations:** ICU, intensive care unit; ARDS, acute respiratory distress syndrome; ACE, angiotensin-converting enzyme; PCR, polymerase chain reaction; WHO, World Health Organization; CRS, cytokine release syndrome; ACEi, angiotensin-converting enzyme inhibitor; ARB, angiotensin receptor blocker; RAS, Renin-Angiotensin System; COPD, Chronic Obstructive Pulmonary Disease; CKD, Chronic Kidney Disease; CVD, Cardiovascular Diseases; CRP, C-Reactive Protein; CD4, Cluster of Differentiation 4; CD8, Cluster of Differentiation 8; LRTI, Lower Respiratory Infection; MODS, multiple organ dysfunction syndromes; MERS, Middle East Respiratory Syndrome; NK cell, Natural Killer cells; SARS-COV, Severe Acute RespiratorySyndrome Coronavirus; SPHCC, the Shanghai Public Health Clinical Centre; T cells, Tymus cell.

presence of comorbidity, diabetes, hypertension, fever, shortness of breath, having oxygen saturation (SPO2) less than 93%, cold, fatigue, ward of admission, the severity of COVID-19 disease, duration of treatment as well as the use of remdesivir, Actemra® and Gammaraas®. Most patients recovered after receiving a combination treatment (oseltamivir + remdesivir + Antibiotics + Vitamin/Mineral) for approximately 11 days with a 90% survival rate. On the contrary, patients who received oseltamivir + hydroxychloroquine + Gammaraas® + antibiotics +Vitamin/Mineral, had a 83% survival rate after being admitted to the hospital for about ten days.

## Conclusion

Factors influencing the development of a cytokine storm include age, duration of treatment, comorbidity, symptoms, type of admission ward and severity of infection. Most patients (76.92%) with cytokine storm who received Gammaraas®/Actemra®, survived although they were in the severe and critical levels (87.17%). Overall, based on the treatment duration and survival rate, the most effective therapy was a combination of oseltamivir + favipiravir + hydroxychloroquine + antibiotics + vitamins/minerals.

## Introduction

Cytokine storm, also known as hypercytokinemia or "cytokines release syndrome," is implicated in the worsening of several conditions, including pulmonary inflammation leading to acute respiratory distress syndrome (ARDS) and in severe COVID-19 cases. Nevertheless, there is lack of information on the latter to provide a better understanding of its role clinically. Information on the impact of cytokine storm in elucidating inflammation, especially involving treatments and vaccines given to the patients is also lacking. The mortality rate from COVID-19 acute respiratory distress is reported to be almost 40–50% [1, 2].

Cytokine storm has been implicated in the 2005 outbreak of the avian H5N1 influenza virus, where patients went on to have a high fatality rate. Cytokine storms might explain why some people have a severe reaction to COVID-19 while others experience only mild symptoms. Such reason may also explain why younger people are less affected where it produces lower levels of inflammation due to less involvement of the immune system [3].

Most currently published papers evaluate COVID-19 patients based on the severity and complications. In this study, the focus is on unvaccinated COVID-19 patients with cytokine storm who received treatment in a private hospital in Indonesia. It is hoped that the data allows easy comparison of the potential impact of cytokine storms among hospitalized patients. Overall, the days of hospitalization, bleeding complications, complications leading to ARDS, secondary bacterial infection and multiple organ failure among COVID-19 may all impact patients with cytokine storm [4, 5].

The SARS-CoV2 spike protein causes cellular infection by tightening the angiotensin-converting enzyme (ACE)-2 on human cells [6, 7]. Cellular infection and viral replication impact the activation of inflammasomes in the host cell [7–10]. The phenomenon will lead to the release of pro-inflammatory cytokines and cell death by pyroptosis with the subsequent release of a damage-associated molecular pattern which reinforces the inflammatory response [7–10].

Antiviral treatment may play a role in the management of COVID-19, especially in more severe disease where immunomodulatory treatments blunting cytokine release may be useful

when appropriately timed [7]. Usually, patients with cytokine storms have fever which may be of high grade in severe cases [11, 12]. In addition, patients have fatigue, anorexia, headache, rash, diarrhea, arthralgia, myalgia and even neuropsychiatric findings. These symptoms may 1) be directly due to the cytokine-induced tissue damage or acute-phase physiological changes or 2) result from immune–cell-mediated responses [11]. Cases can rapidly lead to disseminated intra-vascular coagulation with either vascular occlusion or catastrophic hemorrhages, dyspnea, hypox-emia, hypotension, hemostatic imbalance, vasodilatory shock as well as death [11]. Many patients have respiratory symptoms including cough and tachypnea that can manifest in an acute respira-tory distress syndrome (ARDS), with hypoxemia requiring mechanical ventilation [11]. Addition-ally, renal failure, acute liver injury or cholestasis and a stress-related illness like cardiomyopathy can also occur in severe cases of cytokine storm [11, 13]. Nevertheless, to our knowledge, no study has focused on treatment outcome of cytokine storm among COVID-19 patients.

## Materials and methods

### Study design and setting

The study only involved positive COVID-19 patients before the process of vaccination in Indo-nesia based on the treatment guidelines by the Ministry of Health Indonesia. The study was conducted in a private hospital in West Jakarta between March and September 2020. Only patients who received medical therapy (antiviral-antibiotics) for COVID-19 and met the inclu-sion criteria below were included. This is a cohort retrospective study conducted to support the association between the suspected cause and disease, with pre-existing risk factors and dis-ease with the study variables viewed through historical patient records. A convenient sampling was used to select the patients who fulfilled the inclusion criteria.

### Selection criteria

Inclusion criteria

1) Patients aged $\geq$ 18 years old.

2) COVID-19 positive patients (with or without comorbidities).

3) COVID-19 positive patients who received COVID-19 treatment (antiviral-antibiotic) therapy between March and September 2020.

Exclusion criteria

1) Patients below 18 years.

2) Patients who do not have complete medical record data, being referred or about to be discharged.

### Ethical approval

The research was approved by a local institutional ethics committee at the Faculty of Health, Esa Unggul University with reference number: 0374–20.362/DPKE-KEP/FINAL-EA/UEU/ XII/2020 which complies with the Declaration of Helsinki.

### Clinical outcome parameters

In this study, recovery and death are two categories of the clinical outcome defined. The recov-ery parameter was based on the polymerase chain reaction (PCR) test results for COVID-19

from either the nose/throat/airway aspirate swabs. The findings must be negative at least twice in a row within more than 24 hours. The death parameter is a further positive confirmatory case of COVID-19, where patients died during admission. Treatment outcome was based on the duration of treatment, which is equal to the length of hospital stay.

### Data collection and analysis

The data based on the patient's medical record was gathered using a research data collection sheet containing the patient's identity, medical history, type of treatment therapy and clinical status and all data were fully anonymized. Before collecting the data, the letter approval of research from the hospital and ethical approval from the ethical committee was required. Due to the retrospective study, informed consent from the patients was not necessary with the condition all these data are confidential. All those things had been discussed with the management hospital and ethical committee. Subsequently, the data was entered into the Microsoft Excel 2013 program and was processed using an SPSS version 26.0 program. Data analysis was conducted using Fisher's Exact, Chi-Squared, Independent T-Test, Mann Whitney U and survival test analyses (Kaplan Meier Survival Analysis).

The staging process was done on admission. Staging is based on the Decree of the Minister of Health of the Republic of Indonesia [14], where patients deemed to be in stage 1 had mild symptoms and patients in stage 2 had moderate symptoms. Patients in stage 3 were those with severe symptoms while patients in stage 4 were patients who had to be admitted to the intensive care unit (ICU) for ventilators. Overall, there were only thirteen ventilators available in the hospital, which were all situated in the ICU for patients with oxygen saturation (SpO2) below 93%. Patients were treated in the hospital until the PCR swab was negative.

## Results and discussion

To our knowledge, the impact of cytokine storm on COVID-19 patients has not been clinically described elsewhere. This is the first study to describe the impact of cytokine storms on COVID-19 patients prior to the vaccination process that took place in Indonesia. Currently, COVID-19 infection remains as an unprecedented challenge across the globe when the World Health Organization (WHO) declared it as a global pandemic on March 11, 2020 [15]. The disease can cause severe illness including cytokine storm, leading to hospitalization, ICU admission and even death, especially in older patients (elderly) with underlying extreme health conditions [16].

Based on the patient's medical records, 124 patients who were COVID-19 positive and received treatment in a Private Hospital in West Jakarta during the study duration were recruited. From this number, only 91 (73.4%) patients met the inclusion-exclusion criteria and were included in the study. Among these patients, 14.3% of patients had cytokine storms (Table 1). Most were males (61.5%) with a mean age of 60 years. All patients who had cytokine storm experienced hypertension with diabetic complications (100%).

Fever, cough and shortness of breath were the symptoms commonly appearing among patients with cytokine storm (100.0%). Except for a single patient (7.7%) who had a mild-moderate symptom, almost all cytokine patients with severe and critical illnesses were treated in the ICU. A significant correlation was established between cytokine storms with age, the presence of comorbidity, diabetes, hypertension, fever, shortness of breath, SPO2 < 93%, cold, fatigue, type of admission ward as well as the severity of COVID-19 disease (p < 0.05).

Most (76.9%) patients who had cytokine storm received hydroxychloroquine, but all patients' were given antibiotics and vitamin/mineral (Table 2). The antibiotics were a five days regimen of azithromycin, followed by levofloxacin (seven days). However, critically ill patients

**Table 1. Correlation between cytokine storm, sociodemographic and clinical symptoms.**

| Variable | Frequency (%) | | | p-value |
|---|---|---|---|---|
| | Overall (n = 91) | Cytokine storm (n = 13) | No cytokine storm (n = 78) | |
| **Mean age** | **49.0 ± 16.4** | **61.6 ± 13.0** | **46.9 ± 16.0** | **0.002**[#] |
| Gender | | | | 0.517[*] |
| Male | 64 (70.3) | 8 (61.5) | 56 (71.8) | |
| Female | 27 (29.7) | 5 (38.5) | 22 (28.2) | |
| **Presence of comorbidity** | **49 (53.8)** | **13 (100.0)** | **36 (46.2)** | **< 0.001**[**] |
| Type of comorbidity | | | | |
| **Diabetes** | **28 (30.8)** | **9 (69.2)** | **19 (22.4)** | **0.002**[*] |
| **Hypertension** | **28 (30.8)** | **8 (61.5)** | **20 (25.6)** | **0.019**[*] |
| Symptoms | | | | |
| **Fever** | **62 (68.1)** | **13 (100.0)** | **49 (62.8)** | **0.008**[*] |
| Cough | 62 (68.1) | 13 (100.0) | 49 (62.8) | 0.596[*] |
| **Shortness of breath** | **43 (47.3)** | **13 (100.0)** | **30 (38.5)** | **< 0.001**[**] |
| **SPO$_2$ (< 93%)** | **23 (25.3)** | **12 (92.3)** | **11 (14.1)** | **< 0.001**[*] |
| **Cold** | **30 (33.0)** | **0 (0.0)** | **30 (38.5)** | **0.004**[*] |
| **Fatigue** | **29 (31.9)** | **0 (0.0)** | **29 (37.2)** | **0.008**[*] |
| Anosmia | 11 (12.1) | 0 (0.0) | 11 (14.1) | 0.354[*] |
| Sore throat | 10 (11.0) | 0 (0.0) | 10 (12.8) | 0.347[*] |
| Diarrhoea | 8 (8.8) | 0 (0.0) | 8 (10.3) | 0.596[*] |
| Nausea and vomiting | 8 (8.8) | 0 (0.0) | 8 (10.3) | 0.596[*] |
| **Admission ward** | | | | **< 0.001**[*] |
| Intensive care unit (ICU) | 23 (25.3) | 12 (92.3) | 11 (14.1) | |
| Non-intensive care unit (Non-ICU) | 68 (74.7) | 1 (7.7) | 67 (85.9) | |
| **Severity of COVID-19 disease** | | | | **< 0.001**[*] |
| Mild | 48 (52.7) | 0 (0.0) | 48 (61.5) | |
| Moderate | 20 (22.0) | 1 (7.7) | 19 (24.4) | |
| Severe | 10 (11.0) | 9 (69.2) | 1 (1.3) | |
| Critical | 13 (14.3) | 3 (23.1) | 10 (12.8) | |
| Cytokine storm | | | | |
| Yes | 13 (14.3) | | | |
| No | 78 (85.7) | | | |

[*]Fisher exact test;

[**]Chi Square test;

[#]Independent T-test.

were administered with meropenem until the regimen was completed. A plasma immunoglobulin (gammaraas®) and/or tocilizumab (interleukin-6 receptor antagonist; actemra®) injection were administered to ameliorate the cytokine storms. Additionally, a significant correlation was established between the cytokine storm with the duration of treatment as well as the use of remdesivir, actemra® and gammaraas® (p <0.05).

There was a male predominance among patients with cytokine storms. Based on the data by the WHO, due to their reduced immune function and compensated lungs, smokers have the highest risk of COVID-19 [17]. Other studies [18, 19] reported that smokers had worse outcomes following SARS-CoV-2 viral infection which may explain the higher predominance among males having cytokine storms in our study since the percentage of male smokers in Indonesia was 68.1% as reported in 2016 [20].

**Table 2. Correlation between cytokine storm experienced with duration and type of treatment.**

| Variable | Frequency (%) | | | p-value |
|---|---|---|---|---|
| | Overall (n = 91) | Cytokine storm (n = 13) | No cytokine storm (n = 78) | |
| **Median duration of treatment in days** | **14.0 (5–39)** | **25.0 (7–35)** | **14.0 (5–39)** | **0.001#** |
| Type of treatment | | | | |
| Oseltamivir | 61 (67.0) | 8 (61.5) | 53 (67.9) | 0.752* |
| Antibiotics | 84 (92.3) | 13 (100.0) | 71 (91.0) | 0.587* |
| Vitamin/Mineral^ | 91 (100.0) | 13 (100.0) | 78 (100.0) | 1.000* |
| Hydroxychloroquine | 60 (65.9) | 10 (76.9) | 50 (64.1) | 0.531* |
| **Remdesivir** | **11 (12.1)** | **4 (30.8)** | **7 (9.0)** | **0.048*** |
| Favipiravir | 6 (6.6) | 1 (7.7) | 5 (6.4) | 1.000* |
| **Actemra®** | **5 (5.5)** | **5 (38.5)** | **0 (0.0)** | **< 0.001*** |
| Chloroquine | 4 (4.4) | 0 (0.0) | 4 (5.1) | 1.000* |
| Azithromycin | 7 (7.7) | 0 (0.0) | 7 (9.0) | 0.587* |
| Gammaraas® | **8 (8.8)** | **8 (61.5)** | **0 (0.0)** | **< 0.001*** |

*Fisher exact test;

#Mann Whitney U-test.

^ Vitamins used were vitamins B-complex and C; the type of mineral administered was zinc.

Based on our data, almost all patients who had cytokine storms were treated in the ICU indicating their poorer prognosis. In contrast, the study by Cappanera et al. reported the presence of cytokine storm in only 52% patients in stage III (critical illness). It has been reported that cytokine storm which is an overreaction of the body's immune system is fatal and may be present not only in COVID-19 patients but also in influenza [21]. In fact, cytokine storm is deemed as the primary determinant of the pathophysiological worsening of COVID-19 [22].

In terms of age, there was a higher preponderance of cytokine storm in the elderly COVID-19 patients (61.6 ± 13.0 years). Similarly, Leandri et al., reported that SARS-CoV-2 infection which causes strong immune system dysfunction as characterized by a marked increase in pro-inflammatory response in the host and a life-threatening cytokine release syndrome (CRS) can induce acute respiratory syndrome (ARDS), especially in aged people [23]. Furthermore, the worsening of anti-inflammatory responses in the elderly may be attributed to the higher involvement of the pro-inflammatory responses [24]. In our study, approximately 50% of the COVID-19 patients with cytokine storm had hypertension while the remaining had diabetes. The finding is not surprising since it has been reported that inflammation, oxidative stress and endothelial dysfunction are associated with biological aging and progress in hypertension [25]. As for diabetes mellitus, type 2 diabetes mellitus is primarily reported to be linked to both oxidative stress and ageing [26]. Similarly, hypertension and diabetes mellitus are significant risk factors associated with death among COVID-19 in Mexico [27] where these diseases are the two main comorbidities among COVID-19 patients [17] in that country, besides in China.

Most patients received angiotensin-converting enzyme inhibitor (ACEi) and an angiotensin receptor blocker (ARB) as antihypertensives [28]. Nevertheless, it has been reported that the first step of SARS-CoV-2 infection in humans occurs via viral engagement with cell-surface ACE2 [29] which ultimately affects ACE2 tissue expression or activity. Therefore, further investigation is required to justify whether the influence of Renin-Angiotensin System (RAS) inhibitors on COVID-19 is beneficial, unremarkable, or harmful [29]. Additionally, hyperglycemia as seen in diabetes mellitus also contributes to immune response dysfunction, causing a poorer recovery from various pathogens (bacterial, viral or fungal infections) [30].

Besides hypertension and diabetes, comorbidities such as Chronic Obstructive Pulmonary Disease (COPD), Chronic Kidney Disease (CKD) and Cardiovascular Diseases (CVD) are also significant risk factors in the development of severe COVID-19 disease [31]. Similarly, Otuonye et al. reported that out of the 154 patients studied in Lagos, Nigeria, there was a 2.6% mortality rate occurring among patients with hypertension, diabetes and respiratory tract infections. Lower Respiratory Infection (LRTI) and pneumonia are reported to aggravate the disease further [32]. Generally, the more comorbid a person is (for example, having diabetes and heart disease), the higher are the chances for severe complications from COVID-19 [33].

Only a few studies have investigated the effect of the longer duration of hospitalization on cytokine storms experienced in COVID-19 patients although treatment duration is also generally influenced by the patient's immune system and the severity of the disease. In a study by Chen et al. on patients (n = 249) treated at the Shanghai Public Health Clinical Centre (SPHCC), 86.3% were found to have recovered and were discharged after 16 days of hospitalization [34]. Based on several other studies in China, the average length of treatment is approximately 10 to 13 days [16, 35]. Nevertheless, hospitalization is dependent on various factors associated with COVID-19 where reduced immune systems and higher comorbidities are often seen in the elderly [36] which will inevitably affect the length of treatment as well as the clinical outcome.

In this study, prevalence of cytokine storm was influenced by age, comorbidity, symptoms, length of treatment as well as the treatment received (remdesivir, Gammaraas® and actemra®). Fang, Karakiulakis and Roth's earlier reported that both age (the elderly) and the presence of comorbidities including hypertension and diabetes account for the higher mortality rate due to COVID-19 [37]. Although COVID-19 affects almost all ages, available data suggests the fact that the elderly, besides presenting a higher risk of both getting infected and severe complications, represent 80% of hospitalizations and pose a 23-fold greater threat of death [38]. Overall, age and comorbidities impact disease severity through immune-mediated mechanisms since they are correlated with a chronic rise in pro-inflammatory mediators that cause an enhanced susceptibility to induce an immune dysregulation following SARS-CoV-2 infection [39] as also reported by Fiorino et al. [40].

In this study, cytokine storms were also significantly correlated with fever, shortness of breath and having an SPO2 of less than 93%. One of the most severe complications of COVID-19 is the development of an atypical upper respiratory tract pneumonia that poses a significant challenge for clinicians when managing the disease where abnormal and uncontrolled production of cytokines have been observed in critically ill patients with COVID-19 pneumonia [41]. In fact, fever, cough, shortness of breath and fatigue were the initial symptoms of most patients with SARS COV-2 with some patients even developing chest distress, myalgia and rhinorrhea [42].

The days of hospitalization observed in this study was approximately 25 days. Another study in Malaysia reported that the length of hospitalisation in both the ward and the ICU for COVID-19 patients were only 1–5 days (51.8%), 6–10 days (10.9%) and 11 days (37.3%) [43]. It is plausible that the discrepancies are attributed by the different staging of the disease where most Malaysian patients suffered from stages 1 and 2. On the other hand, Nadeem et al. reported that the median length of hospital stay in Dubai was 19 days, ICU stay was 14 days and patients on ventilator 11 days [44]. Overall, most patients who experienced cytokine storms were treated in the ICU and the poorer prognosis contributed to the long duration of treatment [7].

The type of treatment that a patient receives also affects the clinical outcome of hospitalized patients with cytokine storms. Currently, patients with COVID-19 disease frequently use available therapeutic drugs based on the patient's symptomatic condition, antiviral therapy,

antibiotics, systemic corticosteroids and anti-inflammatory drugs (including anti-arthritis medications) [45]. Based on the Ministry of Health Republic Indonesia guideline, these patients received antiviral (oseltamivir, favipiravir, remdesivir, hydroxychloroquine or chloroquine), antibiotic (azithromycin or levofloxacin) and vitamin such as vitamins C (500 mg) and B complex and minerals [46]. This guideline is in line with the COVID-19 treatments approved by WHO [47] with the addition of the vitamins and the antimalarials.

One of the signs and symptoms of COVID-19 disease is pneumonia where azithromycin or levofloxacin are commonly instituted [48]. Usually, as recommended by the WHO, pneumonia is reported in COVID-19 patients who are at stage 2 and above [49] as well as in hospitalised patients to avoid complications. In a study conducted by Wang et al., at the Union Hospital, Wuhan, most patients received antiviral and antibiotic therapies [50]. Based on the data on 1,099 laboratory-confirmed COVID-19 patients from 552 hospitals in 30 provinces in China, most patients (58.0%) received intravenous antibiotic therapy while 35.8% received oseltamivir treatment [51]. Another study conducted by Ramatillah & Isnaini in Indonesia indicated that patients receiving the Oseltamivir + Hydroxychloroquine combination had the highest survival rate at about 83% after undergoing treatment for approximately ten days [51].

In this study, 14.29% of COVID-19 patients with cytokine storm were administered with Actemra® or Gammaraas®. Tocilizumab, an IL-6 inhibitor, is deemed as relatively effective and safe for cytokine storms [52]. Additionally, corticosteroids, programmed cell death protein (PD)-1/PD-L1 checkpoint inhibition, cytokine-adsorption devices, intravenous immunoglobulin and antimalarial agents have potentially been reported to be beneficial against cytokine storm in COVID-19 patients [52]. Nevertheless, there is a report that the timing of treatment and the proper selection of patients are also key to the success of the therapy against cytokine storm [53].

Based on Table 3, only 23.1% of patients died due to cytokine storms while the remaining 76.9% of patients recovered. However, no correlation was established between cytokine storm and the treatment outcomes.

A survival analysis was conducted to determine the patient's survival rate following COVID-19 treatment. Based on Fig 1, COVID-19 patients who received the combination treatment (Oseltamivir + Hydroxychloroquine + antibiotics + Vitamin/Mineral) had the highest survival rate (approximately 90%) following 11 days of treatment. However, patients who received the combination chloroquine + antibiotics + Vitamin/Mineral had the lowest survival rate (about 22%) after 11 days.

Patients with cytokine storm received several combination of treatments 1) Hydroxychloroquine + Actemra® + Antibiotics + Vitamin/Mineral 2) Oseltamivir + Hydroxychloroquine + Gammaraas® + Antibiotics + Vitamin/Mineral 3) Oseltamivir + Hydroxychloroquine +- Actemra® + Antibiotics + Vitamin/Mineral 4) Remdesivir + Actemra® + antibiotics + Vitamin/Mineral or 5) Favipiravir + Gammaraas® + Antibiotics + Vitamin/Mineral. However, patients who received Oseltamivir + Hydroxychloroquine + Gammaraas® + antibiotics

Table 3. Correlation between cytokine storm experienced and treatment outcomes.

| Variable | Frequency (%) | | | p-value |
|---|---|---|---|---|
| | Overall (n = 91) | Cytokine storm (n = 13) | No cytokine storm (n = 78) | |
| **Treatment outcomes** | | | | 0.389* |
| Death | 13 (14.3) | 3 (23.1) | 10 (12.8) | |
| Recovered | 78 (85.7) | 10 (76.9) | 68 (87.2) | |

*Fisher exact test.

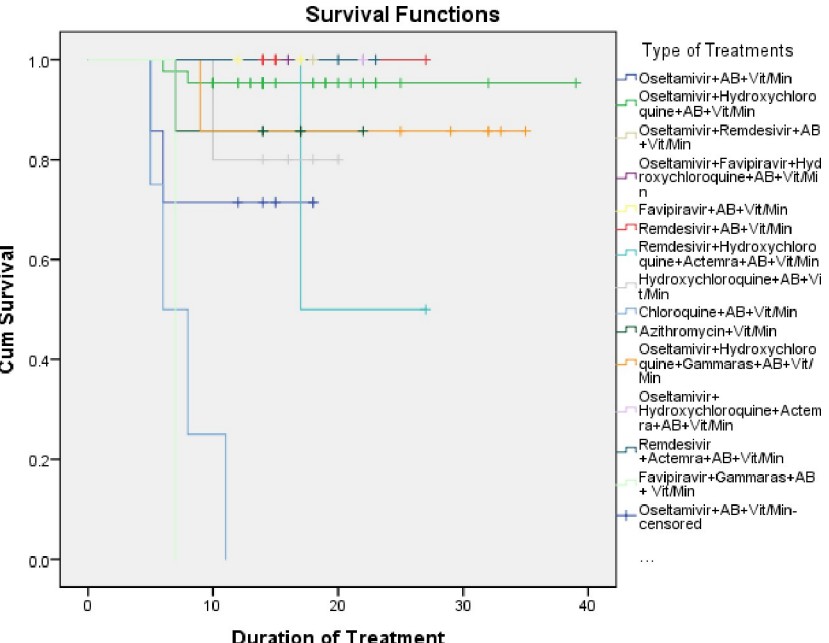

**Fig 1.** *Kaplan meier* survival for COVID-19 patients based on the medical treatments received.

+Vitamin/Mineral had the highest survival rate (83%) following a 10 days hospital admission (Fig 1).

Cytokine storm is the main factor contributing to COVID-19 exacerbation or even death, not only because of pulmonary injury, but also the subsequent development of extrapulmonary multiple-organ failure [54, 55]. Cytokine storms of viral origin showed similar pathogenesis of having an imbalanced immune response with the exaggerated inflammatory reaction in combination with the reduction as well as functional exhaustion of T-cells [54].

Besides COVID-19, cytokine storms are also present in influenza, SARS, and MERS. Based on the immunopathology in COVID-19, SARS, and influenza infections, all the diseases show some hyperinflammation, hypercytokinemia, increased CRP levels, hyperferritinemia associated with poor prognosis, immunosuppression, lymphopenia associated with the severe disease, reduction of NK cell counts in the blood and/or NK cell dysfunction and reduction in both CD4 and CD8 T-cell counts in the blood associated with more severe disease. However, MERS tends to reveal hyperinflammation and hypercytokinemia associated with a poorer prognosis [54].

In our study, most (70%) COVID-19 patients with cytokine storms survived. Sun et al. reported that most patients developed mild symptoms among 41 patients involved in their research [56]. In contrast, some patients developed very severe symptoms and eventually died due to multiple organ dysfunction syndromes (MODS) as a result of a severe cytokine storm [56].

The causes of mortality among COVID-19 patients with cytokine storms were disease severity and the presence of comorbidities. Since Jakarta is the capital city with vast population, suppressing the spread of COVID-19 remains a big challenge for the government. In fact, till date, Jakarta remains as a red zone for COVID-19 [57]. When this study was conducted, the percentage of COVID-19 infected was very high [57], affecting the number of available rooms at the hospital [57] and causing delayed treatment in the hospital. Overall, these circumstances lead to patient admission at the late stages and poorer prognosis.

The study has some limitations since the study duration is rather short and the fact that the published care data to date, originates from a small observational data (no more than 250 patients). Additionally, the treatment type administered is not based on the WHO guideline but rather, on the local context in Indonesia, besides the fact that it was conducted prior to vaccine introduction in the country. Further research should focus on cytokine storms experienced in vaccinated patients to determine if vaccination improves the prognosis of patients with cytokine storms further. Additionally, since our data reported mainly on males and also in the elderly, it would be interesting to see the incidence in females as well as in the younger age group and their responses against cytokine storms (Fig 2).

## Conclusions

All patients with cytokine storm had hypertension or/ and diabetes complications. The age, duration of treatment, comorbidity, symptoms, ward of admission and severity of disease all are important contributing factors to the development of the cytokine storm. Although most patients with cytokine storm (87.17%) were at severe and critical levels of COVID-19, most (76.92%) survived following administration of Gammaraas®/ Actemra®. Overall, based on the treatment duration and survival rate, the most effective therapy was a combination of oseltamivir + favipiravir + hydroxychloroquine + antibiotics + vitamins/minerals.

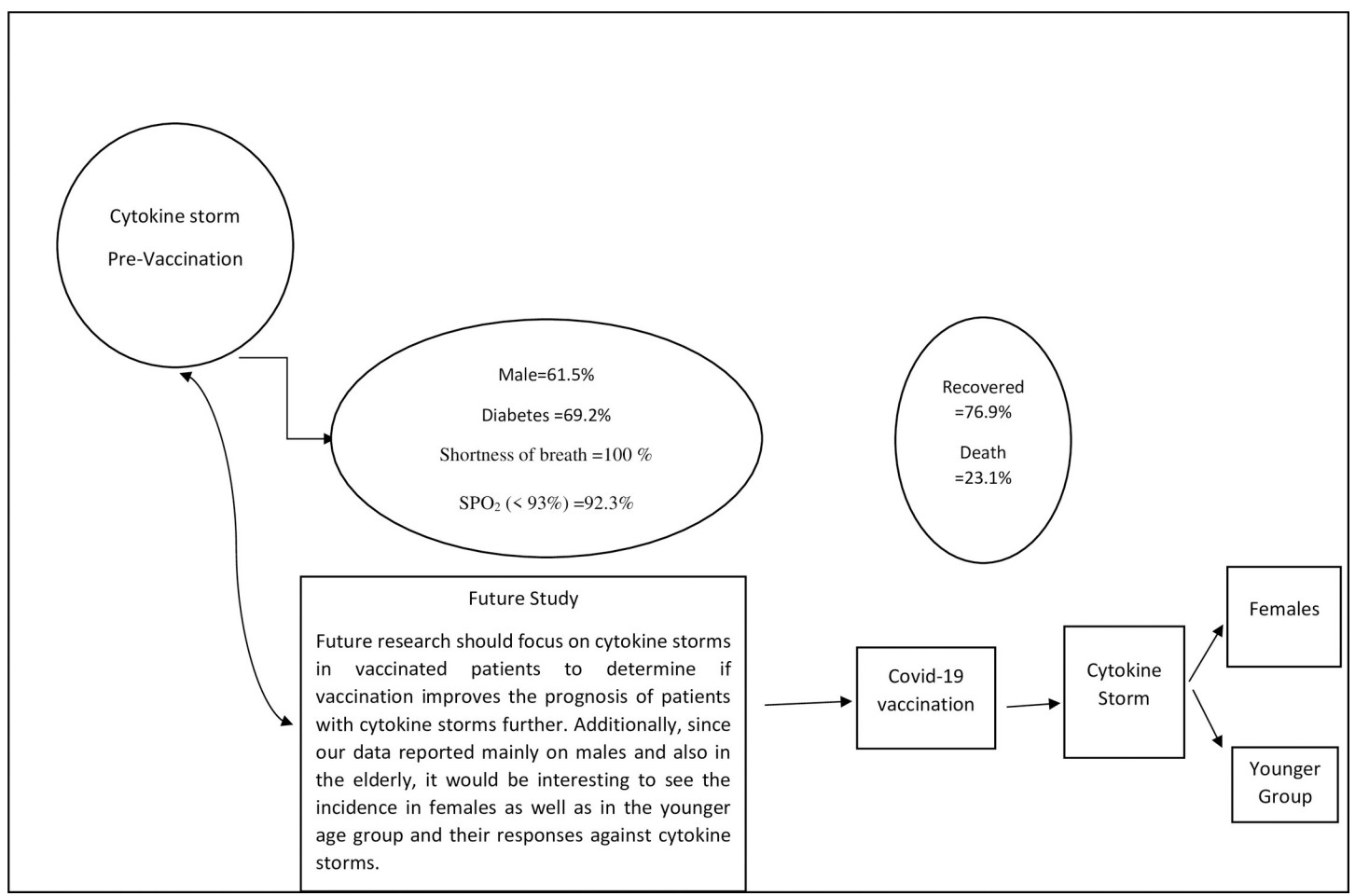

**Fig 2. Summary of the study and future study.**

## Acknowledgments

We would like to acknowledge the Ministry of Education and Research Indonesia (Kemenristek Dikti) and the Education fund management agency (LPDP Indonesia) for their support under the World Professor Class (WPC) programme.

## Author Contributions

**Conceptualization:** Diana Laila Ramatillah.

**Data curation:** Diana Laila Ramatillah.

**Formal analysis:** Ammar Ali Saleh Jaber, Usman Abu Bakar.

**Investigation:** Ika Pratiwy.

**Methodology:** Diana Laila Ramatillah, Usman Abu Bakar.

**Project administration:** Ika Pratiwy.

**Resources:** Ika Pratiwy.

**Supervision:** Siew Hua Gan, Syed Azhar Syed Sulaiman.

**Validation:** Diana Laila Ramatillah, Siew Hua Gan, Syed Azhar Syed Sulaiman, Nina Jusnita, Stefanus Lukas.

**Visualization:** Siew Hua Gan, Syed Azhar Syed Sulaiman, Ammar Ali Saleh Jaber, Nina Jusnita, Stefanus Lukas, Usman Abu Bakar.

**Writing – original draft:** Diana Laila Ramatillah.

**Writing – review & editing:** Siew Hua Gan.

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
