## [Decision Letter · Decision Letter 0]

9 Dec 2021

PONE-D-21-31980Impact of Cytokine Storm on severity of COVID-19 infection in a private Hospital in West Jakarta Prior to VaccinationPLOS ONE

Dear Dr. Ramatillah,

Thank you for submitting your manuscript to PLOS ONE. After careful consideration, we feel that it has merit but does not fully meet PLOS ONE’s publication criteria as it currently stands. Therefore, we invite you to submit a revised version of the manuscript that addresses the points raised during the review process.

We look forward to receiving your revised manuscript.

Kind regards,

Sanjay Kumar Singh Patel, Ph.D.

Academic Editor

PLOS ONE

Journal Requirements:

We would like to acknowledge Ministry of Education and Research Indonesia (Kemenristek Dikti) and Education fund management agency (LPDP Indonesia) for the support under the World Professor Class (WPC) programme.

The author(s) received no specific funding for this work. Only the funded to held the WCP programme not for publication

Reviewers' comments:

Reviewer's Responses to Questions

**Comments to the Author**

1. Is the manuscript technically sound, and do the data support the conclusions?

Reviewer #1: Yes

Reviewer #2: Yes

Reviewer #3: Yes

2. Has the statistical analysis been performed appropriately and rigorously? 

Reviewer #1: Yes

Reviewer #2: Yes

Reviewer #3: Yes

3. Have the authors made all data underlying the findings in their manuscript fully available?

Reviewer #1: Yes

Reviewer #2: Yes

Reviewer #3: Yes

4. Is the manuscript presented in an intelligible fashion and written in standard English?

Reviewer #1: Yes

Reviewer #2: Yes

Reviewer #3: Yes

5. Review Comments to the Author

Reviewer #1: The research article entitled, "Impact of Cytokine Storm on severity of COVID-19 infection in a private Hospital in West Jakarta Prior to Vaccination" by Ramatillah et al., assessed the role of cytokine storms in patients’ high mortality rate during COVID-19 pandemic using retrospective cohort study included only COVID-19 positive patients hospitalized in a Private Hospital in West Jakarta between March and September 2020. Authors found the clinical outcome of COVID-19 patients following medical therapy was either cured (85.7%) or died (14.3%), with 14.3% patients reported to have cytokine storm, from which 23.1% led to fatalities. Finally, authors have concluded that based on the treatment duration and survival rate, the most effective therapy was a combination of oseltamivir+favipiravir+hydroxychloroquine+antibiotics+vitamins/mineral. Altogether this is an important and timely research article, this reviewer has certain suggestions that would help produce a more comprehensive overview of the topic:

Comments:

1. Did authors find any global data on female COVID-19 patients with cytokine storms? This will be noteworthy to know and authors can include this data to their study.

2. Did authors/or any other find cytokine storms in relatively younger COVID-19 patients and how was the response to the cytokine storms?

3. At least one supplementary Figure as illustration may be afforded as to highlight the summary or prospect of this study.

4. Author may explain how to adjust the limitation in their study?

5. The abbreviations should be cross validated in the manuscript (First define them fully followed by abbreviation) and one paragraph can be added for abbreviations.

Reviewer #2: This retrospective study investigated the impact of cytokine storm on severity of COVID-19 disease in a private Hospital in West Jakarta Prior to Vaccination. The authors should distinguish infection from disease, because in COVID-19 disease, SARS-CoV-2 infection severity (high viral load) is not corelated to COVID-19 disease severity [1]

In Title, “COVID-19 infection” should be “COVID-19 disease”

Page 5, line 5, “severity of COVID-19 infection” should be “severity of COVID-19 disease”

Page 6, line 12 from bottom, “more severe infection” should be “more severe disease”

Page 9, line 6, “the severity of COVID-19 infection” should be “the severity of COVID-19 disease”

Page 9, Table 1, row 24, “Severity of COVID-19 infection” should be “Severity of COVID-19 disease”

Page 11, line 3 from bottom, “COVID-19 infection” should be “COVID-19 disease”

Page 12, line 6, “following COVID-19 infection” should be “following SARA-CoV-2 viral infection”

Page 14, last line, “patients with COVID-19 infection” should be “patients with COVID-19 disease”

Page 15, line 9, “COVID-19 infection” should be “COVID-19 disease”

Page 16, Conclusion, line 2, “severity of infection” should be “severity of disease”

Reference

1. Shah S, Singhal T, Davar N, Thakkar P. No correlation between Ct values and severity of disease or mortality in patients with COVID 19 disease. Indian J Med Microbiol. 2021; 39(1): 116- 117.DOI: 10.1016/j.ijmmb.2020.10.021

Reviewer #3: In this paper entitled "Impact of cytokine storm on the severity of COVID-19 infection in a private hospital in West Jakarta prior vaccination", the authors used the convenience sampling method and included subjects who are not vaccinated. The clinal outcomes of the study are fascinating. The manuscript showed that cytokine storm was more in males (mean age 60) than women and interestingly, all hypertension/diabetes subjects experienced cytokine storm. It has many exciting findings which make this manuscript engaging and lucid. However, It has minor concerns.

Minor comments:

1) Please add limitations of the study.

2) The cytokine storm is a fatal reaction of human immune system. Cytokine storm is present in influenza, SARS, and MERS. In few lines, please compare it. (doi:10.1016/j.clim.2020.108652).

---

## [Author Response · Author response to Decision Letter 0]

20 Dec 2021

Reviewer 1

1. Did authors find any global data on female COVID-19 patients with cytokine storms? This will be noteworthy to know and authors can include this data to their study.

Answer : Unfortunately, to date, there is no reported global data on female COVID-19 patients with cytokine storm, to date, since our data is the first reported one on male.

2. Did authors/or any other find cytokine storms in relatively younger COVID-19 patients and how was the response to the cytokine storms?

Answer : Similar to point 1 above, since our sample population was mostly among the older age group (mean age of 60 years), we could not report on the incidence of cytokine storms in relatively younger COVID-19 patients as well as their responses. It would be interesting to see the differences in a future study, using our data as a basis for the comparison.

3. At least one supplementary Figure as illustration may be afforded as to highlight the summary or prospect of this study.

Answer : We have included a supplementary Figure accordingly. Refer to Figure 2.

4. Author may explain how to adjust the limitation in their study?

Answer : This is something that is challenging since they are indeed the limitations of the study. Nevertheless, we have attempted to explain it more clearly in the text now and have added another statement: “Additionally, since our data reported mainly on males and also the elderly, it would be interesting to see the incidence in females as well as in the younger age group and their responses against cytokine storms”. Thank you for the suggestion. 

5. The abbreviations should be cross validated in the manuscript (First define them fully followed by abbreviation) and one paragraph can be added for abbreviations.

Answer: We have added it in the newly-revised manuscript. Thank you for the suggestion.

Reviewer 2

Reviewer #2: This retrospective study investigated the impact of cytokine storm on severity of COVID-19 disease in a private Hospital in West Jakarta Prior to Vaccination. The authors should distinguish infection from disease, because in COVID-19 disease, SARS-CoV-2 infection severity (high viral load) is not correlated to COVID-19 disease severity [1]

In Title, “COVID-19 infection” should be “COVID-19 disease”: This has been changed accordingly, thank you.

Page 5, line 5, “severity of COVID-19 infection” should be “severity of COVID-19 disease”. This has been changed accordingly, thank you.

Page 6, line 12 from bottom, “more severe infection” should be “more severe disease”. This has been changed accordingly, thank you.

Page 9, line 6, “the severity of COVID-19 infection” should be “the severity of COVID-19 disease”. This has been changed accordingly, thank you.

Page 9, Table 1, row 24, “Severity of COVID-19 infection” should be “Severity of COVID-19 . This has been changed accordingly, thank you.

disease”

Page 11, line 3 from bottom, “COVID-19 infection” should be “COVID-19 disease”. This has been changed accordingly, thank you.

Page 12, line 6, “following COVID-19 infection” should be “following SARA-CoV-2 viral infection”. This has been changed accordingly, thank you.

Page 14, last line, “patients with COVID-19 infection” should be “patients with COVID-19 disease”. This has been changed accordingly, thank you.

Page 15, line 9, “COVID-19 infection” should be “COVID-19 disease”. This has been changed accordingly, thank you.

Page 16, Conclusion, line 2, “severity of infection” should be “severity of disease”. This has been changed accordingly, thank you.

Answer: We have added all in the newly-revised manuscript. Thank you for the suggestion.

Reviewer 3

Reviewer #3: In this paper entitled "Impact of cytokine storm on the severity of COVID-19 infection in a private hospital in West Jakarta prior vaccination", the authors used the convenience sampling method and included subjects who are not vaccinated. The clinal outcomes of the study are fascinating. The manuscript showed that cytokine storm was more in males (mean age 60) than women and interestingly, all hypertension/diabetes subjects experienced cytokine storm. It has many exciting findings which make this manuscript engaging and lucid. However, It has minor concerns.

Thank you for the overall positive comments on our paper. We highly appreciate it.

Minor comments:

1) Please add limitations of the study.

Answer : We have added the limitation of the study in the manuscript accordingly

The study has some limitations since the duration is rather short and the fact that the published care data to date came from a small observational data (no more than 250 patients). Additionally, the types of treatment administered are not based on the WHO guideline but rather, on the local context, besides being conducted during the time when vaccines have not been introduced in the country. We have elaborated further to include the fact that future study should focus on females and the younger age group since our sample includes mostly males and the elderly.

2) The cytokine storm is a fatal reaction of human immune system. Cytokine storm is present in influenza, SARS, and MERS. In few lines, please compare it. (doi:10.1016/j.clim.2020.108652).

Thank you very much for the excellent suggestion. We have added a paragraph “Besides COVID-19, cytokine storms are also present in influenza, SARS, and MERS. Based on the immunopathology in COVID-19, SARS, and influenza infections, all the diseases show some hyperinflammation, hypercytokinemia, increased CRP levels, hyperferritinemia associated with poor prognosis, immunosuppression, lymphopenia associated with the severe disease, reduction of NK cell counts in the blood and/or NK cell dysfunction and reduction in both CD4 and CD8 T-cell counts in the blood associated with more severe disease. However, MERS tends to reveal hyperinflammation and hypercytokinemia associated with a poorer prognosis [56].” and have also quoted the said paper accordingly. Thank you.

---

## [Editor Report · Decision Letter 1]

24 Dec 2021

Impact of Cytokine Storm on severity of COVID-19 disease in a private Hospital in West Jakarta Prior to Vaccination

PONE-D-21-31980R1

Dear Dr. Ramatillah,

We’re pleased to inform you that your manuscript has been judged scientifically suitable for publication and will be formally accepted for publication once it meets all outstanding technical requirements.

Kind regards,

Sanjay Kumar Singh Patel, Ph.D.

Academic Editor

PLOS ONE

---

## [Editor Report · Acceptance letter]

31 Dec 2021

PONE-D-21-31980R1 

Impact of Cytokine Storm on severity of COVID-19 disease in a private Hospital in West Jakarta Prior to Vaccination 

Dear Dr. Ramatillah:

I'm pleased to inform you that your manuscript has been deemed suitable for publication in PLOS ONE. Congratulations! Your manuscript is now with our production department. 

Kind regards, 

on behalf of

Dr. Sanjay Kumar Singh Patel 

Academic Editor

PLOS ONE